# Liouna: Biologically Plausible Learning for Efficient Pre-Training of Transferrable Deep Models

Fady Rezk [1]    Antreas Antoniou [1]    Henry Gouk [1]    Timothy Hospedales [1,2]

## Abstract

Biologically plausible learning algorithms, inspired by the inherent constraints of biological neural systems, offer a promising path towards communication and memory-efficient learning with extreme parallelizability where layers learning is decoupled to train in parallel. In this work, we introduce Liouna (Arabic for "plasticity"), an unsupervised biologically plausible local learning algorithm inspired by predictive coding and masked image modelling. We derive Liouna's update rule, which elegantly reduces to a simple Hebbian rule with subtractive inhibition. We establish new state-of-the-art results for local learning rules across CIFAR-10, CIFAR-100, STL-10, and Imagenette, without imposing training procedures that hinder the attainability of the true benefits of local learning. Remarkably, we discover and demonstrate an emergent behaviour in Liouna, where it learns inter-class similarity and separability through feature sharing and specialization, despite observing no labels during training. Notably, we are the first to study the transfer performance of local learning algorithms. By pre-training on unlabelled data, Liouna outperforms previous state-of-the-art methods on 6 out of 8 downstream tasks and even surpasses end-to-end (E2E) supervised training in the low compute regime. Liouna also demonstrates competitive performance with SimCLR pre-trained models in the resource-limited pre-training scenario. This highlights Liouna's potential for efficient transfer learning and/or acceleration of the initial stages of pre-training improving its convergence rates in wall-clock time.

[1]School of Informatics, University of Edinburgh, Edinburgh, UK [2]Samsung AI Research Center, Cambridge, UK. Correspondence to: Fady Rezk <s1985200@ed.ac.uk>.

Accepted to the Workshop on Advancing Neural Network Training at International Conference on Machine Learning (WANT@ICML 2024).

## 1. Introduction & Related Work

Backpropagation (backprop) is de facto the algorithm for training deep neural networks. It uses a task-specific global objective and the chain rule to attribute errors of the objective to specific neurons in the network. Meanwhile, local learning rules (LLRs) are training algorithms that only use information local to a layer, such as inputs and outputs, to update the layer's parameters in building deep representations (Illing et al., 2021; Miconi, 2021; Halvagal & Zenke, 2023). This facilitates extreme parallelization by decoupling the training of different layers, potentially enabling the training of all layers in parallel on neuromorphic devices (Journé et al., 2023; Miconi, 2021). Conversely, backprop is update-locked, meaning the computation of updates is delayed until both forward and backward passes across all layers are processed, limiting the vertical parallelization of backpropagation. Furthermore, all information required for the backprop update, such as intermediate activation values, must be stored in memory, limiting its utility for resource-constrained on-device learning (Zhao et al., 2022). Local learning rules enable memory-efficient on-device learning and reduced communication overhead when models are distributed across devices, complementing their inherent efficiency advantages over backprop.

As the field converges toward training few large-scale multi-modal foundational models on huge datasets with tremendous compute and fine-tuning-based personalization (Yin et al., 2024), improving the efficiency of learning algorithms is paramount for improving time-to-result. By extension, this enables the iterative development of foundational models. Reducing memory burden is a necessity for on-device adaptation and widespread deployment of foundation models.

One class of communication and memory-efficient learning, and therefore parallelizable methods are those that are biologically plausible such as LLRs. Biological systems are inherently constrained in aspects that backprop does not satisfy. These constraints include local plasticity, where updates depend only on local information to a neuron such as inputs and outputs as formalized by Hebbian theory (Hebb, 2002). Moreover, synaptic changes occur based on the activity of neighbouring neurons, rather than requiring global in-

formation transfer and the coordination of distant processing units. This local nature of updates aligns with how the brain processes information and adapts, avoiding the weight transport problem (Crick, 1989) and update locking as present in backprop. Moreover, backprop is temporally non-local limiting it's implementation on neuromorphic devices which are the main contender addressing von-Neumann architecture bottlenecks since backprop hinders achieving both local memory storage and computation using the same nodes as occurs in synapses (Frenkel et al., 2023). For temporal faithfulness, the network should either model time-based spiking activity with feedback connections (Nunes et al., 2022) or instead only be allowed to use local eligibility traces through recurrent/lateral connection of recent local activity (Gerstner et al., 2018).

Previous work on LLRs includes block-wise methods that divide the network into blocks trained separately with backprop (Pyeon et al., 2021; Belilovsky et al., 2020; Siddiqui et al., 2024; Löwe et al., 2020; Wang et al., 2023), feedback alignment (Lillicrap et al., 2016; Frenkel et al., 2021; Laborieux et al., 2021; Payeur et al., 2021; Greedy et al.), contrastive learning approaches like Contrastive, Local And Predictive Plasticity (CLAPP) (Illing et al., 2021), and Local Predictive Plasticity (LPL) (Halvagal & Zenke, 2023), and finally Hebbian Learning (Miconi, 2021; Journé et al., 2023; Grinberg et al., 2019). However, a key issue is that the training procedures employed often limit the promised LLR benefits. Networks are scaled by increasing width instead of depth, per-layer architecture search and hyperparameter tuning is performed, and greedy layer-wise pre-training is used - all introducing overhead that hinders parallelization or diminish their gains. Evaluation procedures are also problematic, with strong inductive biases from tuned architectures potentially inflating reported results attributed to algorithms.

To truly unlock the advantages of LLRs, we propose key desiderata: 1) Locality and weight-transport freedom 2) Update unlocking and temporal locality for parallelization 3) Increasing performance with depth scaling, not width 4) Stability in simultaneously training all layers 5) Being unsupervised to leverage unlabeled data 6) Learning transferable hierarchical representations 7) Robustness across architectures without per-dataset tuning.

We introduce Liouna, a new state-of-the-art biologically plausible LLR inspired by predictive coding framed as a temporal masked image modeling task. We derive Liouna's elegant update rule, showing it reduces to a simple Hebbian form with subtractive inhibition. Liouna establishes new state-of-the-art records across multiple datasets while meeting the desiderata. Remarkably, we are the first to study the transfer capability of LLRs, showing Liouna's pre-trained models outperform previous methods on 6 out of 8

downstream tasks, surpass end-to-end supervised training in the low compute regime and produces competitive results with SimCLR trained models in resource-constrained pre-training scenarios.

## 2. Liouna: Overview and Derivation

We frame predictive coding (Millidge et al., 2022) as a temporal masked image modelling task (Woo et al., 2023b; He et al., 2021) in the representation space without decoding (Baevski et al., 2022; Woo et al., 2023b). The agent goal is to maintain a world model that is predictive of the ground truth state of the world given partial observability. We argue for biological plausibility in Subsection 2.1

Given an image or activations, $\mathbf{x} \in \mathbb{R}^{C \times H \times W}$, we produce corrupted samples $\tilde{\mathbf{x}} = \mathbf{x} \odot \mathbf{m}$, where $\mathbf{m} \in \mathbb{R}^{C \times H \times W}$ are independent Bernoulli random variables with probability $m$ of being zero, and $\odot$ denotes element-wise multiplication. The goal is then to minimize the reconstruction loss $\mathcal{L}^{(l)}$ for layer $l$ as:

$$\mathcal{L}^{(l)}(\mathbf{x}) = \frac{1}{\Omega(\mathbf{x})} \left|\left| f^{(l)}(\mathbf{x}) - f^{(l)}(\tilde{\mathbf{x}}) \right|\right|_1 \qquad (1)$$

where $f^{(l)}(\mathbf{x}) = \mathbf{x}\mathbf{W}^{(l)}$, $W^{(l)}$ is the weights of layer $l$, and $\Omega(\mathbf{x})$ is the total number of elements in the activations. One key issue with this formulation is total collapse. Without further constraints, setting all deep model weights to 0 is in the set of feasible solutions to the minimum of Equation 1. Therefore, we cast the problem as

$$\begin{aligned} \underset{\mathbf{W}^{(l)}}{\arg\min} \ &\mathcal{L}^{(l)}(\mathbf{x}) \quad \forall l \in L \\ \text{s.t. } &||\mathbf{W}_i^{(l)}||_2^2 = c^{(l)}, \end{aligned} \qquad (2)$$

where $\mathbf{W}_i^{(l)}$ is hidden neuron $i \in I$ on layer $l$. This simply enforces a norm constraint on the hidden neurons to be of length $c$. To solve this objective, we make use of the proximal gradient method. The proximal operator for the norm constraint is defined as

$$\text{prox}_g(\mathbf{W}) = \underset{\mathbf{V}}{\arg\min} \left[ \frac{1}{2} |\mathbf{V} - \mathbf{W}|_2^2 + g(\mathbf{V}) \right], \quad (3)$$

where $g(\mathbf{V})$ enforces the constraint $||\mathbf{V}_i^{(l)}||_2 = c^{(l)}$ for each $i$. Specifically, for the norm constraint, $g(\mathbf{V})$ can be represented as an indicator function

$$g(\mathbf{V}) = \begin{cases} 0 & \text{if } ||\mathbf{V}_i^{(l)}||_2 = c^{(l)}, \forall i \\ +\infty & \text{otherwise.} \end{cases} \qquad (4)$$

The proximal gradient update step for the weights $\mathbf{W}^{(l)}$ is given by

$$\mathbf{W}_{k+1}^{(l)} = \text{prox}_g \left( \mathbf{W}_k^{(l)} - \eta \nabla \mathcal{L}^{(l)}(\mathbf{W}_k^{(l)}) \right) \qquad (5)$$

where $\eta$ is the learning rate, and $\nabla \mathcal{L}^{(l)}(\mathbf{W}_k^{(l)})$ is the gradient of the loss function with respect to the weights at iteration $k$. This proximal update ensures that after each gradient descent step, the weights are projected back onto the constraint set defined by $||\mathbf{W}_i^{(l)}||_2 = c^{(l)}$. For $\mathbf{W}_i^{(l)} \neq \mathbf{0}$, the proximal operator $\text{prox}_g$ for the norm constraint can be computed by normalizing each neuron weight $\mathbf{W}_i^{(l)}$ to have an L2 norm equal to $c^{(l)}$,

$$\mathbf{W}_i^{(l)} \leftarrow \frac{c^{(l)}}{||\mathbf{W}_i^{(l)}||_2} \mathbf{W}_i^{(l)}. \tag{6}$$

In the case where $\mathbf{W}_i^{(l)} = \mathbf{0}$, any point with a norm of $c$ is a feasible solution to the proximal operator optimization problem, though we note that this case is unlikely to occur in practice.

## 2.1. Biological Plausibility and Algorithm

A single update requires two views of the data: the original inputs $x$, and masked inputs $\tilde{x}$. To argue for biological plausibility, we adapt a temporally-constrained approach akin to that of CLAPP (Illing et al., 2021). In Hebbian learning rules (Hebb, 2002), the update of a connection $\mathbf{W}_{ji}$ from neuron $i$ to $j$ is allowed to depend only on pre and post-synaptic activities at times $t$ and $t-1$. Information at time $t$ comes from somatic activities while past information can come from recurrent dendritic connections. This is aligned with neuroscientific findings where dendritic and somatic information last for (50-100ms) and (2-10ms) respectively (Major et al., 2013). Somatic and dendritic activities in our context come from partial and full observability of the environment. For biological processing to minimize the effort of predictively coding information being processed, we assume that the network at time $t-1$, takes a full snapshot of the environment $\mathbf{x}$ and a partially observable snapshot $\tilde{\mathbf{x}}$ at time $t$. Using this information, a neural network can *"sanity check"* its own world model.

At time $t-1$, the outputs $\mathbf{y}^l = f^l(\mathbf{x})$ of layer $l$ are recurrently fed back through lateral dendritic connections $\mathbf{W}^{l,R}$ to be available at time $t$. Moreover, at time $t$, somatic activities are produced as $\tilde{\mathbf{y}} = f(\tilde{\mathbf{x}})$. We assume that dendritic activities $\mathbf{y}^l$ influence the weight updates but not the somatic activity $\tilde{\mathbf{y}}$, following the algorithmic reasons and justifications provided in CLAPP (Illing et al., 2021).

Concretely, given a layer, $l$, we have $f(\mathbf{x}) = \mathbf{x}\mathbf{W}$. Please note that we drop the superscript $l$ for brevity. Dendritic activities are computed as $\mathbf{y}^{(t-1)} = f(\mathbf{x}^{(t-1)})$, and $\mathbf{y}^{(t)} = \mathbf{y}^{(t-1)}\mathbf{W}^R$. Similarly, somatic activities are given by $\tilde{\mathbf{y}}^{(t)} = f(\tilde{\mathbf{x}}^{(t)})$. The loss becomes $\mathcal{L} = \left|\left|\tilde{\mathbf{y}}^{(t)} - \mathbf{y}^{(t)}\right|\right|_1$.

The gradient of the loss with respect to the weights is:

$$\frac{\partial \mathcal{L}}{\partial \mathbf{W}} = \underbrace{\text{sgn}(\tilde{\mathbf{y}}^{(t)} - \mathbf{y}^{(t-1)}\mathbf{W}^R)}_{\substack{\text{Subtractive inhibition} \\ \text{of post-synaptic activity}}} \underbrace{\tilde{\mathbf{x}}}_{\text{pre-synaptic activity}} \tag{7}$$

where

$$\text{sgn}(\tilde{y}^{(t)} - y^{(t)}) = \begin{cases} 1 & \text{if } \tilde{y}^{(t)} > y^{(t)} \\ 0 & \text{if } \tilde{y}^{(t)} = y^{(t)} \\ -1 & \text{if } \tilde{y}^{(t)} < y^{(t)} \end{cases}$$

The gradients of the MIM loss when derived locally for a layer neatly result in a Hebbian learning rule with inhibition. The rule is based on a simple correlation between pre and post-synaptic activities. Please note that if dendritic activities are larger than somatic activities, then the lateral connection inhibits the update in the negative direction to become less aggressive in its dependence on partially observed features when coding the environment. On the other hand, if dendritic activities are less than somatic activities, the learning rule will increase the weights to make the dendritic activities more similar to the somatic activities. Thereby, it encourages the network to specialize more on the given feature to produce semantically similar patterns for future inputs.

**Algorithmic implementation:** The key difference between this derivation and the algorithmic implementation is that the generation of partially observable spaces happens at the input level, while, in the algorithm, we produce masks for clean input activations locally at every layer as shown in Algorithm 1 in Appendix A. Masking at the input level once is denoted as "global masking", while somatic activities at the input of every layer are denoted as "local masking". We show that global masking overall does better than local masking in Appendix B. Nevertheless, the reported results in the rest of the paper use local masking. Furthermore, we initialize the lateral connection $\mathbf{W}^R$ as the identity matrix and do not train it.

## 3. Experimental Setup

We evaluate our proposed Liouna algorithm against several baselines: SoftHebb (previous SOTA) (Journé et al., 2023), Latent Predictive Learning (LPL) (Halvagal & Zenke, 2023), and CLAPP (Illing et al., 2021). We train on image classification tasks using the CIFAR-10 (Krizhevsky, 2012), CIFAR-100 (Krizhevsky, 2012), STL-10 (Coates et al., 2011), and Imagenette (FastAI, 2019) datasets.

For CIFAR datasets, we use 3-block convolutional network architectures. For the more complex STL-10 and Imagenette, we scale to deeper 5 and 6 block networks respectively. To evaluate algorithms robustness, we build three standard CNN block designs that produce architectural

*Table 1.* CIFAR-10/100 test accuracies across local learning algorithms using a 3-block CNN. We use [†] to denote reported results by the original authors of LPL (Halvagal & Zenke, 2023).

|  | ARCH | #PARAMS | CIFAR-10 | AVG. | CIFAR-100 | AVG. |
|---|---|---|---|---|---|---|
| | OURS | 370K | 67.93% | | 38.93% | |
| LIOUNA | PNB | 370K | 69.19% | **68.21%** | 40.63% | **39.58%** |
| | PNB-T | 370K | 67.51% | | 39.19% | |
| | OURS | 370K | 61.95% | | 34.65% | |
| SOFTHEBB | PNB | 370K | 70.44% | 65.22% | 38.35% | 34.99% |
| | PNB-T | 370K | 63.28% | | 31.96% | |
| LPL[†] | VGG-11 | 9M | 59.40% | 59.40% | - | - |

variations. We define these architectures as: Ours (Conv-LN-GELU-AvgPool), PNB (BN-Conv-TriangleReLU-AvgPool) and PNB-T (BN-Conv-GELU-AvgPool) where BN, and LN are batch and layer norm respectively. PNB is the original design used in SoftHebb (Journé et al., 2023). Details on the architectures can be found in appendix C.

To enable a fair comparison across algorithms, we study the effect hyperparameters on the learning algorithms on on CIFAR-10 and establish reasonable default settings per algorithm for every architecture variant, without per-layer tuning. Using the tuned hyperparameters, we train all models for 50 epochs with SGD and evaluate linear readouts on a validation split for early stopping. More information about HPO routine and yielded hyperparameters can be found in Appendix D.

Finally, we investigate transfer learning behaviour of local learning rules. We pre-train a 25M parameter 6-block model on the unlabeled STL-10 data. We choose the the PNB-T variant because it uses the most common and standard block design as common for example in ResNets (He et al., 2016). We also include an end-to-end pre-trained SimCLR model for reference. SimCLR pre-training recipe can be found in Appendix F. Subsequently, we finetune the pre-trained models for 10K steps on various downstream tasks: (1) medical segmentation on the ACDC dataset (Bernard et al., 2018), (2) medical diagnosis on the Diabetic Retinopathy Detection (DRD) dataset (Dugas et al., 2015), (3) image segmentation on the ADE20K dataset (Zhou et al., 2017), (4) video regression on the iWildCam dataset (Beery et al., 2021), (5) few-shot image classification on the CUB-200 dataset (Wah et al., 2011), (6) species categorization and individual identification on the HappyWhale dataset (Ted Cheeseman, 2022), and (7) fine-grained image classification on the Food-101 dataset (Bossard et al., 2014). We refer to Appendix G for details on the finetuning recipes.

## 4. Empirical Results

### 4.1. Comparative Analysis

**Robustness to Architectural Variants:** First, we show the performance of the considered algorithms on the small tasks of training 3-block CNNs on CIFAR-10 and CIFAR-100 datasets using the hyperparameters reported in Appendix D. We train the three architectural variants as introduced in section 3 and detailed in Appendix C. Results shown in Table 1 indicate that Liouna's performance is comparable across architectures. Nevertheless, SoftHebb's best performance is observed on their architecture variant (PNB). We observe a large drop in performance when swapping the triangle+ReLU activation with GELU (PNB-T) from 70.44% to 63.28% on CIFAR-10 and from 38.35% to 31.96% on CIFAR-100. Furthermore, we observed that SoftHebb struggles to train our architectural block producing the worst performance across designs at 61.95%.

Please note the difference in performance between SoftHebb's reported results (80.3%) (Journé et al., 2023) and those shown in Table 1 for PNB architecture (70.44%) although they use the same block design. We attribute this difference to (a) using fewer number of channels, and (b) constraining per-layer HPs. We do not attribute this to implementation details because we reproduced SoftHebb's results in our code base using their per-layer HPs. Finally, LPL underperforms both Liouna and SoftHebb although their network size is $\sim 350\times$ our networks. In addition, we use SGD optimizer while LPL uses Adam which has $4\times$ more memory overhead. Finally, LPL was trained for 800 epochs while our models are only trained for 50.

**Depth Scaling on Deep Networks:** We train 6-block networks of our architectural variants on Imagenette at resolution 160px using both Liouna and SoftHebb. The 6-block variant contains 25M parameters in total. In Fig. 1, we show test performance when probing layers depth-wise. We also report linear readout performance when probing layers of networks randomly initialized using SoftHebb's scheme vs ours (standard Kaiming initialization (He et al., 2015)).

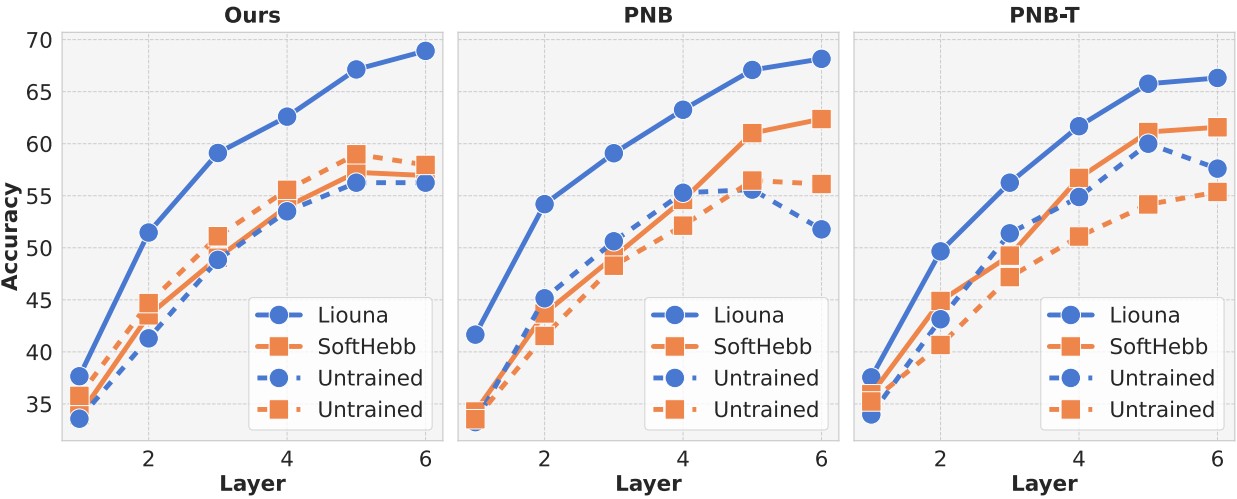

*Figure 1.* Depth-wise linear readout of 6-block CNNs trained on Imagenette. Please note than untrained SoftHebb and Liouna use different initialization schemes.

*Table 2.* Imagenette test accuracies of frozen backbones by the linear readout. Results are for both Liouna and SoftHebb on 6-block CNNs (25M parameters).

|  | ARCH | ACCURACY | | |
|---|---|---|---|---|
|  |  | RANDOM | TRAINED | AVG. |
| LIOUNA | OURS | 56.26% | 68.92% | |
|  | PNB | 51.77% | 68.15% | **67.80%** |
|  | PNB-T | 57.61% | 66.32% | |
| SOFTHEBB | OURS | 57.96% | 56.94% | |
|  | PNB | 56.13% | 62.37% | 60.30% |
|  | PNB-T | 55.36% | 61.58% | |

Final layer test performances are summarized in Table 2. As shown in Fig. 1, Liouna's performance improves by stacking more layers demonstrating successful depth scaling.

Furthermore, with reference to Table 2 and Fig. 1, SoftHebb does worse than random features on our block design. On their block inspired designs (PNB & PNB-T), SoftHebb only manages to provide a marginal improvement over random features. Please note in Fig. 1, that SoftHebb performance gains are achieved in layers 5 and 6 with 1024 and 2048 feature maps implies the necessity for going too wide for the algorithm to learn. Originally, SoftHebb only succeeded in training deep networks by going as wide as having 12,288 feature maps in the final layer, producing 943M parameters in the final layer alone. In contrast, Liouna demonstrates consistent gains in performance with depth scaling across architectural variants. Meanwhile, SoftHebb struggles to both scale to deeper networks and show inconsistent gains across designs.

**Learning Transferable Priors in the Low Labelled Data Regime:** We evaluate Liouna and SoftHebb on STL-10, a benchmark for transferable priors. This dataset offers a small set of labelled data (5K samples) for 10 classes and a large set of unlabeled data (100K samples) with a different distribution. While STL-10 itself uses RGB images, some recent works (e.g., CLAPP) preprocess the data by converting it to grayscale which is an easier task. In contrast, our method directly leverages the richer information present in RGB data, such as textures. This is more complex but holds the potential to lead to representations that generalize better to a wider range of datasets and tasks in real-world applications. Therefore, we reproduce CLAPP's results on RGB images using our low-memory optimizer (SGD) and compute budget (50 epochs).

As shown in Table 3, transferring Liouna's learned prior improves performance over a fully supervised baseline trained without extra data. Our research makes significant progress by being the first to use local methods for learning transferable representations with RGB data. We demonstrate later transfer performance across a wide array of tasks against end-to-end supervised baselines.

On average, Liouna outperforms all baselines in the linear readout regime. LPL and CLAPP did not report results in the finetuning regime neither provided code for computing them. Considering SoftHebb, it is underperforming Liouna with significant margins on all linear readout regimes. For supervised finetuning, results demonstrate a similar brittle phenomenon to those observed on CIFAR-10/100 although more divergent. On average, SoftHebb harms STL-10 transfer learning and E2E supervision in the low data-regime is

*Table 3.* STL-10 test accuracies across local learning algorithms using a 3-block CNN. We report frozen and finetuned backbone. For randomly initialized networks (Random), the Frozen column is linear readout on untrained networks and Finetune columns is end-to-end supervised training. We use * to denote results reproduced with the original authors' codebase of CLAPP (Illing et al., 2021).

| | ARCH | PARAMS | EPOCHS/OPTIM | FROZEN | AVG. | SUPERVISED/FT | AVG. |
|---|---|---|---|---|---|---|---|
| RANDOM | OURS | 6.2M | 50/SGD | 50.83% | | 67.39% | |
| | PNB | 6.2M | 50/SGD | 50.39% | 50.86% | 63.83% | 67.19% |
| | PNB-T | 6.2M | 50/SGD | 51.36% | | 70.36% | |
| LIOUNA | OURS | 6.2M | 50/SGD | 63.30% | | 68.75% (+1.36) | |
| | PNB | 6.2M | 50/SGD | 64.14% | **63.26%** | 66.67% (+2.84) | **69.18%** (+1.28) |
| | PNB-T | 6.2M | 50/SGD | 62.34% | | 72.13% (+1.77) | |
| SOFTHEBB | OURS | 6.2M | 50/SGD | 51.61% | | 63.30% (-4.09) | |
| | PNB | 6.2M | 50/SGD | 60.55% | 55.15% | 72.33% (+8.50) | 67.03% (−0.17) |
| | PNB-T | 6.2M | 50/SGD | 53.31% | | 65.45% (−4.91) | |
| CLAPP* | VGG-6 | 16.2M | 50/SGD | 32.81% | 32.81% | - | - |
| LPL† | VGG-11 | 9.2M | 800/ADAM | 63.20% | 63.20% | - | - |

more beneficial.

### 4.2. Liouna as a Pre-Training Algorithm

We pre-train a 6-block CNN on the unlabeled set of STL-10 for 50 epochs using Liouna's and SoftHebb's appropriate default hyperparameters. We also report results for E2E supervised models and finetuned SimCLR pre-trained model.

**Evidence of Hierarchical Representations - Emergent Clustering Behavior:** The depth scaling observed in Fig. 1 suggests that Liouna is developing hierarchical representations. By visualizing the hidden representations of the Liouna pre-trained model on STL-10, as illustrated in Fig. 2, we discover an emergent behavior wherein Liouna clusters similar concepts while distinguishing dissimilar ones. This clustering phenomenon indicates that Liouna progressively refines feature specialization across conceptual hierarchies. The model learns both discriminative and shared features across categories, effectively aggregating related features while segregating distinct ones. This finding is particularly noteworthy given that Liouna was trained without access to labels, using only a masked image modeling objective. We present additional t-SNE visualizations of hidden activations for STL-10 and Imagenette in Appendix E, specifically in Figures 4 and 3, respectively.

**Transfer Performance of Pre-Trained Models:** With reference to Table 4, we note that Liouna outperforms all supervised baselines showing consistent gains in convergence rates over training from scratch. Given how parallelizable local learning rules are, this demonstrates intriguing potential for turbo-boosting the initial stages of learning. SoftHebb is outperformed by the E2E baseline on 4 tasks out of 8 tasks although it is only trained for 10K iterations. This means

that gains in convergence rates are not consistent; exhibited in only half of the tasks.

Liouna outperforms SoftHebb on 6 out of the 8 considered tasks. Liouna features transfer better in the low data regimes such as CUB-200 amd Food-101. Liouna also exhibits much higher performance on tasks requiring encoding of orientation information such as ACDC, ADE20K, and DRD. On the other hand, SoftHebb features appear to transfer better to regression tasks (iWildCam) and fine-grained identification tasks (HappyWhales identification). Consistent with past findings (Ericsson et al., 2021), there exists no dominant pre-training method across all downstream tasks if the evaluation suite is variable enough.

**Towards Turbo-Boosting Pre-Training:** Our proposed algorithm, Liouna, demonstrates competitive performance across a diverse set of computer vision tasks when compared to state-of-the-art self-supervised learning method, SimCLR. Interestingly, on the ACDC semantic segmentation dataset, Liouna achieves a mean Intersection over Union (mIoU) of 54.52%, close to SimCLR's mIoU of 55.08%. Liouna also exhibits a Mean Absolute Error (MAE) loss of 1.92 on the IWildCam dataset for wildlife camera trap image classification, while SimCLR's MAE loss is 1.85. Furthermore, on the challenging Food-101 and CUB-200 datasets for fine-grained recognition, Liouna attains an impressive top-1 accuracy of 45.00% and 62.75% respectively while SimCLR achieves 48.63% and 63.32% respectively. Notably, by design, the training time for Liouna is dramatically shorter than SimCLR. This observation supports our hypothesis that using Liouna as a warm-up stage before pre-training with SimCLR can dramatically improve convergence rates in wall-clock time. We leave wall-clock gains measurements and the construction of a synergic pre-training scheme be-

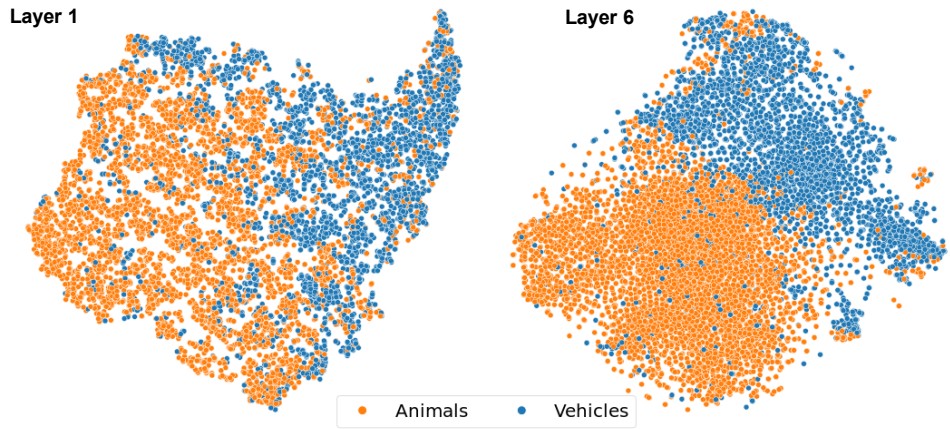

*Figure 2.* Visualizing the STL-10 test set hidden activations of layers 1 and 6 from the pre-trained 6-block CNN (PNB-N). Liouna learns inter-class similarities and dissimilarities. It bring together classes of same type (i.e: vehicles) while pushing apart dissimilar classes (animal and vehicles).

*Table 4.* Finetuning performance performance on our suite of downstream tasks. All models are trained for 10K steps including the end-to-end supervised baseline.

| MODEL NAME | ACDC MIOU↑ | ADE20K MIOU↑ | DIABETIC RET AUC MACRO↑ | FOOD-101 ACC1↑ | HWHALE SPECIES ACC1↑ | HWHALE INDIVIDUAL ACC1↑ | IWILDCAM MAE LOSS ↓ | CUB-200 ACC1↑ |
|---|---|---|---|---|---|---|---|---|
| SIMCLR | 55.08 | 2.90 | 0.72 | 48.63 | 92.82 | 22.37 | 1.85 | 63.32 |
| LIOUNA (FT) | **54.52** | **1.21** | **0.66** | **45.00** | **91.02** | 14.87 | 1.92 | **62.75** |
| SOFTHEBB (FT) | 45.70 | 1.09 | 0.63 | 19.75 | 83.89 | **24.30** | **1.76** | 53.30 |
| E2E SUPERVISED | 28.97 | 1.05 | 0.63 | 39.13 | 87.99 | 11.97 | 2.51 | 58.03 |

tween backpropagation and local learning rules for future work.

## 5. Conclusion & Future Work

This study investigates local learning algorithms as a biologically plausible alternative to backpropagation, aiming to address the limitations of backpropagation that hinder extreme parallelizability and fast training. The key properties of local learning algorithms, such as locality, low communication overhead, and suitability for on-device adaptation, offer advantages over backpropagation. The study critically reviews the literature on local learning algorithms and identifies the training procedures and evaluation protocols that have hindered the realization of these advantages. Based on these findings, the study establishes desiderata for the design and evaluation of local learning algorithms, focusing on their scaling behaviour through deep layer stacking, and suitability for parallelization without preceding pre-training processes of procedural bottlenecks.

Subsequently, we introduced Liouna, a biologically plausible local learning algorithm inspired by masked image

modelling tasks. By evaluating selected baselines against Liouna, we established a new state-of-the-art across the common suite of problems investigated in the biologically plausible local learning literature and demonstrated that Liouna scales gracefully with depth. Notably, to the best of our knowledge, we are the first to study the transfer performance of local learning algorithms. In a suite of downstream tasks, we showed that pre-trained Liouna models outperform the previous state-of-the-art SoftHebb pre-trained models on 6 out of 8 considered tasks. Moreover, we demonstrated that in the low compute regime, Liouna outperforms end-to-end supervised training from scratch, highlighting its potential for efficient transfer learning.

The competitive performance of Liouna compared to Sim-CLR, coupled with its biologically plausible local Hebbian rule and faster convergence, makes it an attractive element to incorporate into self-supervised learning tasks across various computer vision domains. However, to fully unleash the potential of Liouna, further research is needed to explore better training recipes and investigate its scaling laws. Optimizing the training process and understanding the behavior of Liouna at larger scales could potentially lead to even

greater performance gains and efficiency improvements.

The key limitation of the benchmarking of the work introduced in this paper is not reproducing LPL baselines and using CLAPP's original codebase. Nevertheless, we found that LPL underperforms SoftHebb and Liouna by a large margin although it was trained on a larger model with more compute. Therefore, we chose to assign available compute in this study to stronger baselines. Meanwhile, reproducing CLAPP performance plummets when evaluating the algorithm using the original author's codebase with low-memory overhead optimizer (SGD) on a more complex task (RGB images) with a fixed compute budget (50 epochs). As Liouna and SoftHebb show stronger compute efficiency, we maintained SoftHebb as the main contender. Furthermore, local learning rules underperforming backpropagation in the large compute regime remain as an open problem.

Future work must consider scaling laws for local learning algorithms. We believe an understanding of how they scale with data, compute and parameters is key for better pre-training recipes. Furthermore, true extreme parallelizability requires a tremendous amount of infrastructure engineering. By measuring compute scalability using wall-clock time, a more fair comparison against backpropagation can be made and conclusions be drawn. Moreover, a synergic view between the two where local learning is used to accelerate the initial stages of backpropagation training can then be found.

In summary, our results demonstrate the promising capabilities of Liouna as a self-supervised learning algorithm, paving the way for further research and potential adoption in various computer vision and machine learning applications.

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

## A. Algorithm Psuedo-Code

---

**Algorithm 1** Liouna Pseudo-Code

---

**Require:** Model $\mathcal{M}$, input batch $\mathbf{x}$, learning rate $\eta$
 0: **for** each trainable layer $f$ in $\mathcal{M}$ **do**
 0:      $\tilde{\mathbf{x}} \leftarrow \mathbf{x} \odot \text{mask}$
 0:      $\mathbf{y} \leftarrow f(\mathbf{x})$
 0:      $\tilde{\mathbf{y}} \leftarrow f(\tilde{\mathbf{x}})$
 0:      $\mathcal{L} \leftarrow \|\mathbf{y} - \tilde{\mathbf{y}}\|_1$
 0:      $\nabla_{\mathbf{w}_i}\mathcal{L} \leftarrow \frac{\partial \mathcal{L}}{\partial \mathbf{w}_i}$
 0:      **for** each neuron $i$ in layer $f$ **do**
 0:          $\mathbf{w}_i^{(t+1)} \leftarrow \mathbf{w}_i^{(t)} - \eta \nabla_{\mathbf{w}_i}\mathcal{L}$
 0:          $\mathbf{w}_i^{(t+1)} \leftarrow \frac{c^{(f)}}{\|\mathbf{w}_i^{(t+1)}\|_2}\mathbf{w}_i^{(t+1)}$
 0:      **end for**
 0:      $\mathbf{x} \leftarrow \mathbf{x}.\text{detach}()$
 0: **end for**=0

---

## B. Temporal Masking Strategies

According to the derivation of Liouna, the partial observable state comes from partial observablity, or a partial snapshot of the environment. We denote this masking strategy as "global masking". Meanwhile, in implemenation, we mask the pre-synaptic activities at every layer that resulted from the somatic activities of the previous layer. We denote this "local masking". In table TODO, we show that both masking strategies provide equivalent performance

*Table 5.* Linear readout validation accuracy results for 3-layer CNNs trained on CIFAR-10 with different masking strategies.

|  | CIFAR-10 |
| --- | --- |
| **Global Masking** | **67.99%** |
| **Local Masking** | 66.84% |

## C. Architecture Details

. To evaluate the robustness of the selected algorithms, we train three different convolutional network architectures, each employing distinct block designs. These block designs are based on standard practices in well-known architectures (He et al., 2016; Woo et al., 2023a). Please note all convolutional layers preserve resolution and pooling layers downsample by half. First Architecture (Ours) uses a block comprising a convolutional layer, layer normalization, GELU activation, and average pooling. The second architecture (PNB) is inspired by the SoftHebb design. It contains a batch normalization layer, followed by convolutional layer, a triangle+ReLU activation, and average pooling (Journé et al., 2023). Finally, the third Architecture (PNB-T) modifies the SoftHebb block by replacing the triangle+ReLU activation with a GELU activation.

# D. Hyperparameters for SoftHebb and Liouna

We search for algorithm-specific hyperparameters. For SoftHebb, this involves tuning the temperature of the softmax inhibition function. For Liouna, we search for the optimal masking ratio. On small resolution datasets like CIFAR-10/100, we also search for a minimum masking ratio. As the spatial dimensions shrink with increasing depth, high masking ratios can eliminate too much information. Using the maximum and minimum masking ratios, we create a linearly spaced vector of length N, where N is the number of blocks. For instance, if the maximum and minimum masking ratios are 80% and 40%, respectively, the masking ratios for a 3-block network would be [80, 60, 40].

Furthermore, for SoftHebb, their weight initialization and weight-norm dependent learning rate scheduler had hyperparameters that needed tuning, namely initialization radius, and learning rate scheduler power. Nevertheless, we found that SoftHebb's per-layer HPO yielded the same HPs across layers. Therefore, we use these without searching for them.

## D.1. Best HPs

Table 6. Best HPs found for LocalMIM on every arch variant

| LocalMIM | |
|---|---|
| **Our Block Design** | |
| **HP** | **Value** |
| batch size | 1024 |
| LR | 0.01 |
| MR | 80 |
| min MR | 30 |
| SoftHebb (Triangle ReLU) | |
| **HP** | **Value** |
| batch size | 2048 |
| LR | 0.01 |
| MR | 85 |
| min MR | 40 |
| SoftHebb (GELU) | |
| **HP** | **Value** |
| batch size | 1024 |
| LR | 0.005 |
| MR | 80 |
| min MR | 30 |

| SoftHebb | |
|---|---|
| **Global Hyperparameters** | |
| Init Radius | 25 |
| LR Scheduler Power | 0.5 |
| Our Block Design | |
| **HP** | **Value** |
| LR | 0.001 |
| Temp | 1.00 |
| batch size | 256 |
| SoftHebb (Triangle ReLU) | |
| **HP** | **Value** |
| LR | 0.01 |
| Temp | 0.75 |
| batch size | 64 |
| SoftHebb (GELU) | |
| **HP** | **Value** |
| LR | 0.001 |
| Temp | 1.00 |
| batch size | 1024 |

## D.2. Liouna HPO

*Table 7.* HPO. Effect of hyperparameters on LocalMIM trained backbone for 25 epochs. Ours block design

| CIFAR-10 - Batch Size: 512 | |
|---|---|
| For LR experiments, masking ratio is set to 70% | |
| **Learning Rate** | **Accuracy** |
| 0.001 | 65.85% |
| 0.005 | 67.77% |
| **0.01** | **67.94%** |
| 0.05 | 65.84% |
| For masking ratio experiments, LR is set to 0.01 | |
| **Masking Ratio** | **Accuracy** |
| 20% | 65.43% |
| 30% | 65.91% |
| 40% | 67.05% |
| 50% | 68.02% |
| 60% | 67.53% |
| 70% | 67.70% |
| 80% | **68.07%** |
| 85% | 67.705% |
| Masking Ratio and LR set to 80% and 0.01 respectively. Linear probe batch size is fixed to 512. | |
| **Batch Size** | **Accuracy** |
| 64 | 63.71% |
| 128 | 65.94% |
| 256 | 67.34% |
| 512 | 67.73% |
| 1024 | **68.23%** |
| 2048 | 66.84% |
| 4096 | 65.82% |
| Masking Ratio, BS and LR set to 80%, 1024 and 0.01 respectively. A minimum masked ratio is used to produce linear grid of values. Sample L linearly spaced samples in the range [min, max] inclusive. | |
| **Minimum Mask** | **Accuracy** |
| 30% | **68.48%** |
| 40% | 68.11% |
| 50% | 67.98% |
| 60% | 68.14% |

*Table 8.* HPO. Effect of hyperparameters on LocalMIM trained backbone for 25 epochs. SoftHebb block design with Triangle-ReLU

| CIFAR-10 - Batch Size: 512 | |
|---|---|
| For LR experiments, masking ratio is set to 70% | |
| **Learning Rate** | **Accuracy** |
| 0.001 | 66.83% |
| 0.005 | 67.63% |
| **0.01** | **67.52%** |
| 0.05 | 64.31% |
| For masking ratio experiments, LR is set to 0.01 | |
| **Masking Ratio** | **Accuracy** |
| 20% | 67.09% |
| 30% | 67.61% |
| 40% | 67.72% |
| 50% | 68.32% |
| 60% | 68.04% |
| 70% | 67.87% |
| 80% | 68.25% |
| 85% | **68.32%** |
| Masking Ratio and LR set to 85% and 0.01 respectively. Linear probe batch size is fixed to 512. | |
| **Batch Size** | **Accuracy** |
| 64 | 62.64% |
| 128 | 65.56% |
| 256 | 67.48% |
| 512 | 67.63% |
| 1024 | 68.11% |
| **2048** | **68.37%** |
| 4096 | 67.16% |
| Masking Ratio, BS and LR set to 85%, 2048 and 0.01 respectively. A minimum masked ratio is used to produce linear grid of values. Sample L linearly spaced samples in the range [min, max] inclusive. | |
| **Minimum Mask** | **Accuracy** |
| 30% | 68.04% |
| 40% | **68.35%** |
| 50% | 67.61% |
| 60% | 68.32% |

*Table 9.* HPO. Effect of hyperparameters on LocalMIM trained backbone for 25 epochs. SoftHebb Block Design with GELU

| CIFAR-10 - Batch Size: 512 | |
|---|---|
| **For LR experiments, masking ratio is set to 70%** | |
| **Learning Rate** | **Accuracy** |
| 0.001 | 67.16% |
| 0.005 | **67.47%** |
| 0.01 | 67.16% |
| 0.05 | 64.265% |
| **For masking ratio experiments, LR is set to 0.01** | |
| **Masking Ratio** | **Accuracy** |
| 20% | 66.48% |
| 30% | 66.43% |
| 40% | 67.23% |
| 50% | 67.39% |
| 60% | 67.48% |
| 70% | 67.81% |
| 80% | **67.94**% |
| 85% | 67.58% |
| **Masking Ratio and LR set to 85% and 0.01 respectively. Linear probe batch size is fixed to 512.** | |
| **Batch Size** | **Accuracy** |
| 64 | 65.07% |
| 128 | 66.12% |
| 256 | 66.76% |
| 512 | 66.89% |
| 1024 | **67.92%** |
| 2048 | 67.55% |
| 4096 | 67.01% |
| **Masking Ratio, BS and LR set to 80%, 1024 and 0.005 respectively. A minimum masked ratio is used to produce linear grid of values. Sample L linearly spaced samples in the range [min, max] inclusive.** | |
| **Minimum Mask** | **Accuracy** |
| 30% | **68.48**% |
| 40% | 68.11% |
| 50% | 67.98% |
| 60% | 68.14% |

## D.3. SoftHebb HPO

*Table 10.* HPO. Effect of hyperparameters on SoftHebb trained backbone for 25 epochs. Ours block design

| CIFAR-10 - Batch Size: 10 | |
|---|---|
| For LR experiments, batch size and temp are set to 10 and 0.75 | |
| **Learning Rate** | **Accuracy** |
| 0.001 | **62.00%** |
| 0.005 | 60.48% |
| 0.01 | 48.54% |
| 0.05 | 49.73% |
| For Temperature experiments, LR is set to 0.001 | |
| **Temperature** | **Accuracy** |
| 0.25 | 57.69% |
| 0.5 | 58.43% |
| 0.75 | 61.42% |
| 1 | **61.84%** |
| 5 | 56.94% |
| Temperature and LR are set to 0.001 and 1.00 respectively. Linear probe batch size is fixed to 512. | |
| **Batch Size** | **Accuracy** |
| 64 | 59.97% |
| 128 | 61.9% |
| 256 | **62.41%** |
| 512 | 60.1% |
| 1024 | 60.71% |
| 2048 | 60.51% |
| 4096 | 61.85% |

*Table 11.* HPO. Effect of hyperparameters on SoftHebb trained backbone for 25 epochs. SoftHebb block design with Triangle-ReLU

| CIFAR-10 - Batch Size: 10 | |
|---|---|
| For LR experiments, batch size and temp are set to 10 and 0.75 | |
| **Learning Rate** | **Accuracy** |
| 0.001 | 67.57% |
| 0.005 | 70.09% |
| 0.01 | **70.36%** |
| 0.05 | 67.96% |
| For Temperature experiments, LR is set to 0.01 | |
| **Temperature** | **Accuracy** |
| 0.25 | 45.25% |
| 0.5 | 68.36% |
| 0.75 | **70.41%** |
| 1 | 67.60% |
| 5 | 58.54% |
| Temperature and LR are set to 0.01 and 0.75 respectively. Linear probe batch size is fixed to 512. | |
| **Batch Size** | **Accuracy** |
| 64 | **69.91%** |
| 128 | 69.45% |
| 256 | 69.01% |
| 512 | 66.52% |
| 1024 | 64.5% |
| 2048 | 63.81% |
| 4096 | 62.82% |

*Table 12.* HPO. Effect of hyperparameters on SoftHebb trained backbone for 25 epochs. SoftHebb block design with GELU

| CIFAR-10 - Batch Size: 10 | |
|---|---|
| For LR experiments, batch size and temp are set to 10 and 0.75 | |
| **Learning Rate** | **Accuracy** |
| 0.001 | **61.22**% |
| 0.005 | 59.32% |
| 0.01 | 59.18% |
| 0.05 | 48.12% |
| For Temperature experiments, LR is set to 0.001 | |
| **Temperature** | **Accuracy** |
| 0.25 | 49.49% |
| 0.5 | 55.81% |
| 0.75 | 59.95% |
| 1 | **61.89%** |
| 5 | 57.83% |
| Temperature and LR are set to 0.001 and 1 respectively. Linear probe batch size is fixed to 512. | |
| **Batch Size** | **Accuracy** |
| 64 | 60.73% |
| 128 | 60.96% |
| 256 | 61.42% |
| 512 | 62.07% |
| 1024 | **62.5%** |
| 2048 | 61.17% |
| 4096 | 60.98% |

# E. Liouna Hidden Representations Analysis

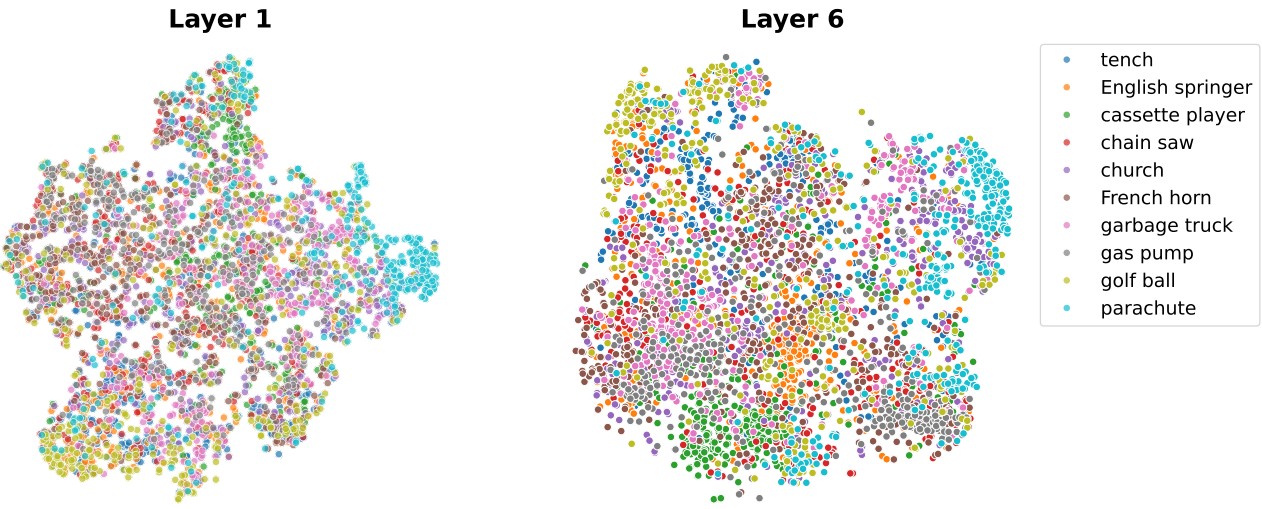

*Figure 3.* Visualizing the imagenette test set hidden activations of layers 1 and 6 from the pre-trained 6-block CNN (ours Block). As imagenette classes have much less in common than STL-10, we see more separability across concepts. For example, tenches, English Springers, and casette players are can be seen to be pushed apart. Meanwhile, a garbage truck and gas pump are pushed closer together.

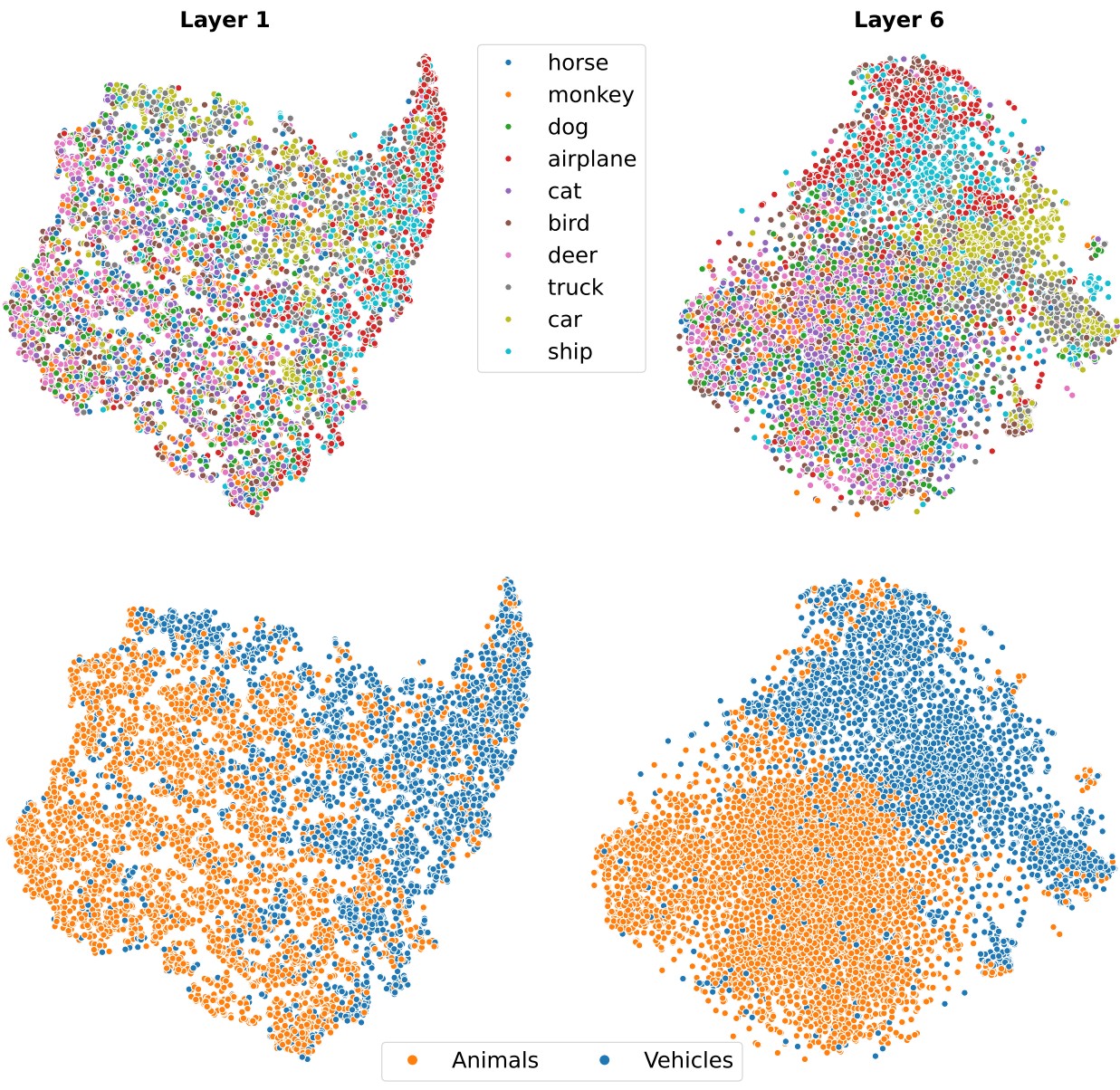

*Figure 4.* Visualizing the STL-10 test set hidden activations of layers 1 and 6 from the pre-trained 6-block CNN (PNB-N). Liouna learns inter-class similarities and dissimilarities. It bring together classes of same type (i.e: vehicles) while pushing apart dissimilar classes (animal and vehicles).

## F. Hyerparameters for SimCLR Pre-Training

This study investigates pre-training in resource-constrained regimes. For reference, we provide pre-trained SimCLR model on the STL-10 dataset. We use SGD optimizer with One-Cycle Learning rate (Smith & Topin, 2018) with max learning rate of 0.01. We train for 50 epochs although its compute in wall-clock time is orders of magnitude longer than all other baselines. We train with a batch size of 256 and termperature of 0.1.

## G. Downstream Tasks Finetuning Recipes

For downstream finetuning, all models are trained for 10K steps and evaluated every 500 itererations. Default optimizer used is AdamW with weight decay value of 0.01. Learning rate scheduler is architecture specific as detailed below. Default learning rate scheduler is plateau annealing with patience of 1000, relative scaling, scale factor of 0.5 and a threshold of 0.0001. Reported results are top-3 validation models (across all validated checkpoint) ensembled by prediction averaging.

Convolution and MLPs only architectures such as classification, recognition and regression tasks uses a learning rate of 1e-3. Meanwhile, segmentation tasks use an 8-block decoder transformer. For the convolution + transformer hybrid, we use a learning rate of 2e-5.

