# OpenReview forum: "Liouna: Biologically Plausible Learning for Efficient Pre-Training of Transferrable Deep Models"
_ICML.cc/2024/Workshop/WANT — WANT@ICML 2024 Poster_

### Official Review · Reviewer_U8yv · 2024-06-13
**Self-supervised local learning rule algorithms show strong performance across a variety of direct and downstream tasks.**

**Confidence:** 3

**Summary:**

This paper proposes Liouna -- a biological-inspired local learning algorithm that self-supervises using masked input samples. Specifically, their algorithm minimizes the mean loss between the representation of standard and masked inputs, using a proximal gradient update to adjust weights as appropriate. Models trained on this method demonstrate superior performance in comparison with similar past algorithms and better scaling and transfer abilities. In addition, the authors propose four key requirements for future local learning algorithms.

**Strengths:**

- The paper is written well and is easy to follow, clearly explaining most biological and algorithmic topics along the way.
- The results demonstrated are strong and extensive across a variety of tasks, from direct classification to transfer learning.
- The authors make a clear case for the efficacy and practicality of their algorithm, and LLRs at large. These include parallel training, efficiency improvements, and reduced memory overhead.
- Tables and graphs are well-formatted and easy to follow.

**Weaknesses:**

- Some equations could be better formatted. For example, $\mathbf{W}$ is never clearly defined in the line after Eq. 1, and $\mathbf{V}$ is not in Eq. 3, which makes the section as a whole harder to follow.
- Despite the topic and subsequent method being interesting and novel, the paper runs relatively short, and some areas that could benefit from being expanded upon are never revisited in the main paper. I will note that the appendix contains a lot of interesting material in this regard, so perhaps it would be beneficial to move some sections there into the main paper.
- Some minor inconsistencies. For example, the authors claim to "discard Softhebb from further consideration" on line 208, yet the remainder of their results significantly rely on more comparisons with the Softhebb algorithm.

**Limitations:**

- The papers brevity given the author's impressive work on an interesting and novel field.

**Suggestions:**

Overall, this was an enjoyable, novel, and well-written paper. Some suggestions include:
- I think the paper could be made a lot stronger simply by adding additional information or elaborating on claims. Some potential suggestions are: background on Softhebb, since it is used extensively during experiments; moving the hidden representation figures in Appendix F into the main paper to further enforce that Liouna shows evidence of hierarchical representations.
- There are some minor typos. For example "Heirarchial" and "heirarchical" on lines 265 and 267 are misspelled. As such, it will be good to comb over the paper a final time.

---

### Official Review · Reviewer_HfKo · 2024-06-14
**New locally learning rule**

**Confidence:** 4

**Summary:**

The paper proposes the novel local learning rule Liouna and demonstrates how it works for transfer learning. Experiments are conducted with CIFAR-10/100, STL-10, and Imagenette datasets.  The proposed approach outperforms alternative LLR methods.

**Strengths:**

1. Extensive experiments and comparisons with competitors are presented
2. A study of hidden representation trained by the proposed approach is shown in Appendix
3. Simple update rule

**Weaknesses:**

1. Some equations look incomplete, e.g., in equations (2) and (3) "min" and "argmin" notation is missed.
2. Notation $W^R$ is unclear; what happens with weight matrix $W$?
3. Algorithm pseudocode is moved to the appendix; this structure complicates understanding the approach
4. Legend in Figure 1 is indistinguishable
5. The obtained accuracy is too low for the current CV networks, which are trained by the backprop algorithm - https://paperswithcode.com/task/image-classification
6. The tested architectures can be easily trained with the backprop, so why LLR is needed here is unclear.
7. The choice of the constraint for the parameters is not well-motivated. There are a lot of other constraints that can be incorporated into the proximal gradient method.

**Limitations:**

Only linear layers are discussed in section 2.1.

**Suggestions:**

1. Improve the quality of presentation, e.g. the motivation for local learning rules is not illustrated in the text and is not even used (e.g. parallelization and asynchronous learning)
2. Test larger models where the vanilla backprop is infeasible or slow
3. Add motivation on the selection this constraint on the parameters per layer.

---

### Meta-Review · Area_Chair_H1Ka · 2024-06-15

**Recommendation:** Accept (Poster)
**Confidence:** 4

**Metareview:**

The paper proposes a biologically plausible local learning algorithm Liouna and demonstrates how it achieves SoTA transfer learning results on image datasets.

Weaknesses noted by reviewers:
* some writing typos and suggestions: (1) some equations look incomplete, (3) unclear notations, (4) the structure can be re-organized, (5) explain the hyper-parameter choice
* Experiments on some challenging tasks.

The AC encourages the authors to thoroughly consider the feedback provided in the individual reviews and use it to enhance the manuscript.

---

### Decision · Program_Chairs · 2024-06-17

**Decision:**

Accept (Poster)

**Comment:**

We thank the authors for their time and contribution to WANT and we are pleased to share that after the reviewing process the paper has been accepted. Congratulations! We encourage the authors to consider reviewers' feedback for the improvement of the camera-ready version. We hope to see you in person at the workshop and brainstorm on efficient training research together!